# Dietary EPA Increases Rat Mortality in Diabetes Mellitus, a Phenomenon Which Is Compensated by Green Tea Extract

**DOI:** 10.3390/antiox8110526

**Published:** 2019-11-04

**Authors:** Thibault Leger, Beibei He, Kasra Azarnoush, Chrystèle Jouve, Jean-Paul Rigaudiere, Florent Joffre, Damien Bouvier, Vincent Sapin, Bruno Pereira, Luc Demaison

**Affiliations:** 1Unité de Nutrition Humaine (UNH), INRA/Université Clermont Auvergne, 63000 Clermont-Ferrand, France; hebeibei0122@hotmail.com (B.H.); kasra.azarnoush@chu-st-etienne.fr (K.A.); chrystele.jouve@inra.fr (C.J.); jean-paul.rigaudiere@inra.fr (J.-P.R.); luc.demaison@inra.fr (L.D.); 2Heart Surgery Department, Gabriel Montpied Hospital, Clermont-Ferrand University Hospital, 63000 Clermont-Ferrand, France; 3Hôpital Nord, Saint-Etienne University Hospital, 42270 Saint-Priest-en-Jarez, France; 4ITERG, 11 rue Gaspard Monge, - ZA Pessac Canéjan, F-33610 Canéjan, France; f.joffre@iterg.com; 5Department of Medical Biochemistry and Molecular Biology, CHU Clermont-Ferrand, 63000 Clermont-Ferrand, France; dbouvier@chu-clermontferrand.fr (D.B.); vsapin@chu-clermontferrand.fr (V.S.); 6Department of Clinical Research and Innovation, CHU Clermont-Ferrand, 63000 Clermont-Ferrand, France; bpereira@chu-clermontferrand.fr

**Keywords:** diabetes, heart, coronary reactivity, EPA, green tea, mitochondria

## Abstract

Diabetes is characterized by a high mortality rate which is often associated with heart failure. Green tea and eicosapentaenoic acid (EPA) are known to lessen some of the harmful impacts of diabetes and to exert cardio-protection. The aim of the study was to determine the effects of EPA, green tea extract (GTE), and a combination of both on the cardiac consequences of diabetes mellitus, induced in Wistar rats by injection of a low dose of streptozotocin (33 mg/kg) combined with a high fat diet. Cardiac mechanical function, coronary reactivity, and parameters of oxidative stress, inflammation, and energy metabolism were evaluated. In the context of diabetes, GTE alone limited several diabetes-related symptoms such as inflammation. It also slightly improved coronary reactivity and considerably enhanced lipid metabolism. EPA alone caused the rapid death of the animals, but this effect was negated by the addition of GTE in the diet. EPA and GTE combined enhanced coronary reactivity considerably more than GTE alone. In a context of significant oxidative stress such as during diabetes mellitus, EPA enrichment constitutes a risk factor for animal survival. It is essential to associate it with the antioxidants contained in GTE in order to decrease mortality rate and preserve cardiac function.

## 1. Introduction

Diabetes is one of the major causes of death in humans, accounting for one death every six seconds worldwide according to the International Diabetes Federation (IDF). The eighth IDF Diabetes Atlas suggests that 425 million people, or one in 11 adults, have diabetes mellitus. This number is predicted to rise to almost 700 million by 2045 (“IDF Diabetes Atlas - 2017” n.d.) [1]. The World Health Organization reports that the annual international cost of diabetes is more than US $827 billion. There are different kinds of diabetes of which diabetes mellitus is the most prevalent. It exists in two major forms: Type I diabetes, characterized by insulin deficiency, and type II, associated with insulin pathway dysfunction. Each form represents 6% and 92% of diabetic people, respectively. Other kinds of diabetes are more specific. These are gestational diabetes, neonatal diabetes, maturity onset diabetes of the young, and latent autoimmune diabetes in adults. Just over 210 million people worldwide are estimated to suffer from undiagnosed diabetes (“IDF Diabetes Atlas - 2017 Atlas” n.d.), including over 90% of those with monogenic diabetes [2].

All forms of diabetes mellitus lead to hyperglycemia [3,4]. Chronic high blood glucose increases HBA1c and promotes end-damage of several organs. Heart failure occurs in more than 30% of diabetic patients [5], although blood glucose control limits the development of complications [6]. Some cardiac alterations appear in the early stage of diabetes mellitus: Impairment of insulin-related pathways, increase in non-esterified fatty acids (NEFAs), lipotoxicity in myocardial tissue, and mitochondrial dysfunctions [7,8]. Mitochondrial dysfunctions cause oxidative stress to increase, leading to complications such as inflammation characterized by the activation of the nuclear factor kappa-B (NF-κB) pathway. This phenomenon stimulates the synthesis of pro-inflammatory cytokines and thus provokes multiple cardiovascular diseases [9,10]. Furthermore, NF-κB induces oxidative stress and increases mitochondrial dysfunctions [11]. Later stage diabetes is characterized by fibrosis with collagen accumulation and coronary microcirculation dysfunctions [12,13]. Diabetic patients often develop electrolyte perturbations such as diabetic ketoacidosis or non-ketotic hyperglycemic hyperosmolar syndrome. This leads to changes in systemic concentrations of potassium, magnesium, phosphate, and calcium ions [14]. Streptozotocin (STZ)-treated rats display reduced myocardial expression of sodium-calcium exchanger 1 (NCX1) calcium channels due to calcium depletion in the blood [15].

The beneficial impact of certain omega 3 (ω3) polyunsaturated fatty acids (PUFAs) in the management of diabetes has recently been highlighted [16,17]. The cardiac benefits of ω3 have recently been the subject of increasing interest. These benefits have been observed during ischemia/reperfusion [18], hypertrophy [19], and arrhythmias [20]. In general, ω3 protection improves the clinical outcomes of cardiovascular diseases [21]. Among the ω3 long-chain PUFAs (LC-PUFAs) of marine origin, eicosapentaenoic acid (EPA) (C20:5 n-3) has particularly noticeable positive effects [22]. Its main cardio-protective property is a result of its anti-inflammatory effects, in contrast with ω6 (omega 6) PUFAs which are pro-inflammatory [23]. Preservation of mitochondrial integrity has also been shown to intervene in the ω3 LC-PUFAs-induced cardio-protection [24]. In the western diet, the ω6/ω3 PUFAs ratio is approximately 15/1. The World Health Organization recommends increasing ω3 dietary intake to 4–5/1 [25]. Lowering this ratio improves cardiac phospholipids’ ω6/ω3 PUFAs ratio, stimulating beneficial effects in patients with cardiac diseases [26]. An improvement in insulin sensitivity has also been observed with ω3 PUFAs in obese subjects [27].

Green tea is also known to reduce risk and mortality due to cardiovascular diseases [28,29]. Catechins, in particular epigallocatechin gallate (EGCG), provide tea with antioxidant and other beneficial properties [30,31]. Its antioxidant properties have been proven in rats subjected to STZ-induced diabetes [32]. In addition, green tea has a hypoglycemic effect during diabetes [33] by improving glucose tolerance and insulin sensitivity in murine models of diabetes [34], reducing hepatic glucose production [35] and limiting intestinal glucose transport [36]. Furthermore, green tea has the potential to inhibit the NF-kB pathway, to limit the release of pro-inflammatory cytokines, and to reduce lipid peroxidation in the hearts of diabetic animals [37,38]. Green tea also has lipid-lowering properties [39]. Finally, Liu et al. recently highlighted the mitigation of cardiac mitochondrial dysfunction in diabetic Goto-Kakizaki rats using EGCG [40].

The purpose of this study was: (i) to determine the impact of diabetes induced by STZ injection and a high-fat diet (HFD) on cardiac activity and mitochondrial function in rats; (ii) to evaluate the effects of enrichments with EPA, green tea extract (GTE), and a combination of both on the cardiac consequences of the diabetes triggered in this study; and (iii) to analyze the mechanisms involved in the observed effects. Several features of myocardial function were assessed: Ex vivo cardiac mechanical function, fatty acid composition of cardiac phospholipids, plasma parameters, and several factors related to oxidative stress and inflammation. 

## 2. Materials and Methods

### 2.1. Ethical Approval

All the experiments followed the European Union recommendations for the care and use of laboratory animals for experimental and scientific purposes. All animal work was approved by the local board of ethics for animal experimentation (Comité d’éthique pour l’expérimentation animale, Auvergne) and declared to our laboratory’s animal research facility (authorization No. APAFIS#14870-2018042711405357v2). The research complied with the ARRIVE animal research guidelines [41].

### 2.2. Experimental Rats and Diet

Fifty three-month-old male Wistar rats (Janvier, Le Genest-Saint-Isle, France) were kept under controlled lighting, humidity, and temperature conditions, with two or three rats per cage. After one week of acclimatization, the rats were randomly allocated to five dietary groups. The control group (CTRL) was fed with a normolipidic diet (A04, Safe, Augy, France). The other groups were subjected to a HFD containing (in % *w*/*w*) casein (27), l-cysteine (0.3), soy oil (3.5), butter (31.7), amylum (5), maltodextrin (8.85), saccharose (13), cellulose (6), mineral mixture (3.5), vitamin mixture (3.5), and choline (0.15). The lipid fraction represented 3% and 35.2% for the normolipidic and high-fat diets. Three diabetic groups were enriched with EPA and/or GTE. In the first group, part of the saturated fatty acids (1.2% of the diet) was replaced by triglyceride-rich EPA. In the second group, part of maltodextrin (0.5% of the diet) was substituted with GTE. Finally, the third group received both substitutes. The animals were fed ad libitum with these different diets for nine weeks. The details of the diets are presented in Table 1. It is notable that no lipid peroxidation occurred in the diets, in spite of the high vulnerability of EPA to damage from reactive oxygen species.

### 2.3. Diabetes Triggering

After three weeks of feeding, the animals of the four groups fed with HFD (diabetes (DIAB), DIAB+EPA, DIAB+GTE, and DIAB+EPA+GTE) were injected intraperitoneally with STZ (33 mg/kg in citrate buffer, pH 4.5) on the 21st day, following an adapted method initially described by Srinivasan et al. [42]. CTRL animals were sham-injected with the same volume of citrate buffer without STZ. 

### 2.4. Morphological Data

Body weight, fat mass, and lean mass were evaluated each week in live animals by magnetic resonance imaging (Echo MRI LL, Houston, TX, USA). Food consumption was measured twice a week in each cage of two or three animals.

### 2.5. Assessment of Ex Vivo Cardiac Mechanical Function 

After the six week post-injection period, hearts were perfused at constant coronary flow using the non-recirculating Langendorff method. A rapid thoracotomy was performed, and the heart was immediately collected and placed in 4 °C saline buffer until cessation of beating. In the first minute after removal, the heart was perfused through the aorta at 37 °C with a Krebs-Heinselett buffer in order to avoid cellular damage and preconditioning problems. The perfusion fluid was composed of (in mM) NaCl (119), KCl (4.8), MgSO_4_ (1.6), NaHCO_3_ (22), KH_2_PO_4_ (1.2), CaCl_2_ (1.8), d-glucose (11), and sodium hexanoate (0.5), pH 7.4. It was maintained at 37 °C and constantly oxygenated with carbogen (95% O_2_, 5% CO_2_). The heart was constantly perfused with a peristaltic pump (Gilson, Middleton, WI, USA), thus upholding the perfusion flow at 14 mL/min until the end of the perfusion. From the fifth minute of perfusion, the hearts were electrically paced at a rate of 368 beats/min. To evaluate cardiac mechanical activity, a latex balloon related to a pressure probe and an amplifier was inserted into the left ventricle. It was inflated until the diastolic pressure reached 10 mmHg. This system meant that systolic, diastolic, and developed pressures as well as heart rate, contraction (dP/dt max), and relaxation (dP/dt min) could be evaluated. The evaluation of these last two parameters was made possible by the use of perfectly standardized perfusion conditions. The coronary pressure was also evaluated with a pressure probe inserted just before the aortic cannula. Changes in the coronary pressure reveal modifications of coronary volume (coronary dilatation, constriction, and/or obliteration of coronary vessels). All the parameters of cardiac function were recorded and analyzed with HSE software (Hugo Sachs Elektronik, March-Hugstetten, Germany).

### 2.6. Coronary Reactivity

At the 50th min of perfusion, the coronary tone was raised by the constant infusion of vasoconstrictor thromboxane analog U46619 (30 nM) into the perfusion system near the aortic cannula, at a maximum rate of 1.5% of coronary flow. This procedure induced a coronary pressure of 95 to 115 mmHg. The vasodilation potential was evaluated with two agents at several doses: Acetylcholine (10, 20, 30, and 40 pmol) and sodium nitroprusside (100, 200, 300, 400, and 500 pmol). Acetylcholine (Ach) and sodium nitroprusside (SN) vasodilatations reflected, respectively, the endothelial-dependent vasodilatation (EDD), due to the simultaneous action of endothelial and smooth muscle cells, and endothelial-independent vasodilatation (EID), due to the relaxation of smooth muscle cells. The dilatation amplitude was calculated as the ratio between the maximal decrease in coronary pressure and the coronary pressure just before the injection of the dilatation agents. As the heart weight and coronary volume were subjected to intra- and intergroup variations, a correction was performed to normalize the input function of vasodilatation agents according to coronary flow. The dose-response curve between the amount of Ach injected and the maximal EDD was then fitted to a third degree polynomial for each heart, thus enabling the performance of statistical analyses. In the case of the EID, the curves were fitted to an Ln function. The vasodilation activities of endothelial cells (ECVA) were also estimated from the corrected EDD and EID. For each heart and each injected Ach dose, the amount of SN (reflecting the amount of dilatation agent) necessary to obtain the same percentage of Ach-induced dilatation was extracted from the EID curve according to the formula: (ECVA) = e^[(% Ach-induced dilatation-b)/a]^, where a and b are the coefficients of the theoretical EID curve. The results were expressed in picomole-equivalents of SN.

At the end of the perfusion protocol, atria and the remaining aorta were cut off from the heart. A piece of myocardium (approximately 200 mg) from the apex of the heart was immediately freeze-clamped and stored at −80 °C for further analysis. The other part of the myocardium was immediately used for mitochondrial preparation.

### 2.7. Preparation of Isolated Mitochondria

The mitochondria were extracted as previously described [43]. Myocardium was shredded with scissors in cold isolation buffer A, composed of mannitol 220 mM, sucrose 70 mM, MOPS 5 mM, and EGTA 2 mM, pH 7.4 at 4 °C, fatty acid-free serum albumin 0.2%. The myocardium pieces were rinsed several times on a filter and homogenized with a blender (0.4 s, rheostat = 3, Polytron) and a Potter-Elvehjem homogenizer containing 15 mL of isolation buffer A. After centrifugation (800× *g*, 10 min, 4 °C), the supernatant with mechanically-extracted mitochondria was kept in ice. The pellet was re-suspended in cold isolation buffer B, composed of KCl 100 mM, MOPS 50 mM, and EGTA 2 mM, pH 7.4 at 4 °C, fatty acid-free serum albumin 0.2%. A protease (subtilisin 0.02%) was added for 1 min to digest myofibrils at ice temperature, and the totality was then homogenized with Potter-Elvehjem homogenizer (300 rpm, three to four transitions). Immediately afterward, subtilisin action was stopped by adding isolation buffer B (30 mL) and centrifuging (8000× *g*, 10 min, 4 °C). The supernatant was discarded and the pellet was re-suspended in isolation buffer B. The homogenate was centrifuged (800× *g*, 10 min, 4 °C), and the resulting supernatant with enzymatically-extracted mitochondria was filtered and collected with the other supernatant. Mitochondria were then washed by centrifuging (8000× *g*, 10 min, 4 °C). The pellet of mitochondria was suspended in cold isolation buffer C, composed of KCl 100 mM, MOPS 50 mM, and EGTA 1 mM, pH 7.4 at 4 °C, re-centrifuged (8000× *g*, 10 min, 4 °C) and finally re-suspended at a concentration of approximately 13 mg/mL. It was then frozen and stored at −80 °C.

### 2.8. Western Blot Analysis

Tissues were ground three times in a mini bead beater in the presence of a lysis buffer constituted of HEPES 50 mM, sodium chloride 150 mM, EDTA 10 mM, anhydrous sodium tetrabasic pyrophosphate 10 mM, β-glycerophosphate 25 mM, sodium fluoride 100 mM, and anhydrous glycerol 1.086 M supplemented with phosphatase inhibitors (Sigma Aldrich, Saint-Quentin-Fallavier, France). Successive centrifugations were performed and the supernatants collected. Protein was quantified using a bicinchoninic acid assay kit (Thermo Fisher Scientific, Asnières-sur-Seine, France). For protein immunoblotting, 25 µg of proteins were loaded for separation by SDS-PAGE electrophoresis and transfer on polyvinylidene fluoride (PVDF) membranes. Membranes were then immunoblotted with the appropriate antibody to detect acetylated-lysine (Ac-K, 1:1000, Cell Signaling #9441), acetylated-superoxide dismutase 2 (Ac-SOD2, 24 kDa, 1:1000, Abcam #ab137037), β-actin (β-actin, 42 kDa, 1 µg/mL, Abcam #ab25894), carnitine palmitoyltransferase 1b (CPT1b, 87 kDa, 1µg/mL, Abcam #ab134988), glutathione peroxidase 1 (GPX1, 22 kDa, 1µg/mL, Abcam #22604), nuclear factor of kappa light polypeptide gene enhancer in B-cell inhibitor alpha (IκBα, 39 kDa, 1:1000, Cell Signaling #9242), NAPDH oxidase 1 (NOX1, 0.5 µg/mL, Abcam #ab131088), pyruvate dehydrogenase acetyl-transferring kinase isozyme 4 (PDK4, 47 kDa, ThermoFisher #PA5-79800), nitrosylated-tyrosine (1:1000, Thermo Scientific #32-1900), sirtuin-1 (Sirt1, 120 kDa, 1:1000, Cell Cignaling #9475), sirtuin-3 (Sirt3, 28 kDa, 1:500, Santa Cruz #sc-365175), superoxide dimustase 1 (SOD1, 18 kDa, 0.2 µg/mL, Abcam #ab13498), superoxide dismutase 2 (SOD2, 22 kDa, 1:1000, Cell Signaling #13194), uncoupling protein-3 (UCP3, 34 kDa, 1:1000, Abcam #ab10985), and voltage-dependent anion-selective channel (VDAC, 32 kDa, 1:1000, Cell Signaling #4866). Antibody binding was detected using horse radish peroxidase (HRP)-conjugated secondary antibodies and ECL Western blotting substrate (Thermo Fisher Scientific, Asnières-sur-Seine, France). 

Immunoblots were visualized using a chemiluminescence imaging system (MF ChemiBIS, DNR bio-imaging systems, Jerusalem, Israel) and quantified using MultiGauge V3.2 software (FSTV, Courbevoie, France). For detection of nitrosylated proteins, protein extraction was performed according to the manufacturer’s instructions. Protein carbonylation was evaluated with a commercially available OxyBlot Protein Oxidation Detection Kit (S7150, Sigma Aldrich, Saint Quentin Fallavier, France). The assessments were performed either on myocardial tissue or on isolated mitochondria. Myocardial and mitochondrial proteins were referred to β-actin and VDAC for their intergroup stability.

### 2.9. Gene Expression

Total ribonucleic acid (RNA) was extracted from 50 mg of cardiac powder using TRIzol^®®^ (Thermo Fisher Scientific, Asnières-sur-Seine, France) according to the manufacturer’s instructions. Chloroform was added (0.2 mL/mL of TRIzol^®®^) and samples were mixed and centrifuged for 15 min at 12,000× *g* at 4 °C. Aqueous phase containing RNA was collected, mixed with isopropanol to precipitate RNA, and centrifuged (12,000× *g*, 4 °C, 15 min). After centrifugation, the pellet was washed with ethanol 70% (*v/v*), and dried and suspended in water. RNA quantification was verified by measuring the ratio of optical densities at 260 nm and 280 nm, whilst RNA integrity was confirmed by agarose gel migration. cDNAs were synthesized from 2 µg of total RNA using the High Capacity cDNA Reverse Transcription Kit from Applied Biosystem (Thermo Fisher Scientific, Asnières-sur-Seine, France). The products of reverse transcription were used for quantitative real-time polymerase chain reaction (qRT-PCR) using specific primers and Rotor-Gene SYBR Green PCR master mix on a Rotor-Gene Q system (Qiagen, Courtaboeuf, France). Messenger RNA (mRNA) was quantified using the standard curve of native cDNA and serial dilutions. The cardiac mRNA expressions were determined for catalase, interleukins 10, 1 β, and 6 (IL10, IL-1 β and IL6), sodium-calcium exchanger 1, sodium-hydrogen antiporter 1 (NHE1), and superoxide dismutase 2. Primer sequences and PCR conditions are presented in Table 2. After testing several housekeeping genes (cardiac α-actin, β-actin, glyceraldehyde-3-phosphate dehydrogenase (GAPDH), 60S acidic ribosomal protein P0 (RPLP0), and hypoxanthine phosphoribosyltransferase 1 (HPRT1)), GAPDH was chosen for its intergroup stability.

### 2.10. Other Biochemical Determinations

During the dietary period, glycemia was determined three times a week in a blood droplet collected at the extremity of the tail with a glucose analyzer (ACCU-CHEK Softclix, Newpharma, Liège, Belgium). Insulinemia was determined on the day of sacrifice using an immunoassay kit (Eurobio Laboratoire Abcys, Courtaboeuf, France). Circulating lipids (non-esterified fatty acids, triglycerides, total, high-density lipoprotein, and low-density lipoprotein cholesterols) were assayed with commercial kits (Diasys, Grabels, France). Other plasma compounds, namely creatinine, sodium, and chloride, were analyzed using commercially-available kits (Abcam, Paris, France). Myocardial total collagen was assessed with a kit manufactured by Quickzyme (Leiden, The Netherlands).

### 2.11. Fatty Acid Composition of Cardiac Phospholipids 

Total lipids were extracted from cardiac tissues according to Folch et al.’s procedure [44]. Lipid extracts were prepared from approximately 50 mg of cardiac tissue using 4 mL of chloroform-methanol (2:1 *v/v*) and 1 mL of 0.9% NaCl. After phospholipid separation on Sep-Pack cartridges and methylation of total lipids, fatty acid methyl esters were separated by gas chromatography as previously described [45].

### 2.12. Statistical Analysis

Results are expressed as mean ± SEM according to statistical distribution (assumption of normality assessed with the Shapiro-Wilk test). Survival curves were treated with a Log-rank Mantel-Cox test. The repeated measures were analyzed by Repeated-Measures ANOVA. Concerning non-repeated measures, quantitative variables were compared between groups by ANOVA or Kruskal-Wallis tests when assumptions required for the ANOVA were not met (normality and homoscedasticity analyzed with the Bartlett test). When appropriate (omnibus *p*-value < 0.05), a post-hoc test to take into account multiple comparisons was performed: Tukey-Kramer post ANOVA and Dunn after Kruskal-Wallis test. When only two groups were compared, a Mann-Whitney test was carried out. Statistical analyses were performed with the NCSS 2010 software (NCSS LLC, Kaysville, UT, USA).

## 3. Results

### 3.1. Glycemia and Insulinemia

All diabetic animals displayed hyperglycemia on or before the third day after the STZ injection (+234% compared to the CTRL rats, *p* < 0.001, Figure 1A). In parallel, insulinemia was reduced in all the STZ-injected rats (−55% compared to the CTRL animals, *p* < 0.05, Figure 1B). The glycemia was then kept at a high level in the diabetic groups, and the insulinemia remained low until the end of the protocol. For these two parameters, no fortification-related differences were observed for the diabetic animals.

### 3.2. Survival Rate

Diabetes induction affected the survival rate of the animals (Figure 2). No death occurred in the CTRL group. However, mortality occurred in all the animals which became diabetic. Statistical analysis of the survival curves indicated that mortality increased greatly in the DIAB+EPA group (no survival at the completion of the protocol) compared to the CTRL group. Addition of GTE alone reduced death compared to the DIAB+EPA group. Furthermore, GTE enrichment greatly prevented the deleterious influence of EPA. However, GTE and GTE+EPA enrichments did not improve survival compared to the DIAB group. Since death occurred in all diabetic groups, the laboratory’s animal wellbeing facility did not authorize the reiteration of the protocol. Taking into account misses due to heart perfusion, the sample sizes at the end of the experiment were eight, five, zero, seven, and five for the CTRL, DIAB, DIAB+EPA, DIAB+GTE, and DIAB+EPA+GTE groups, respectively. 

### 3.3. Fatty Acid Composition of Cardiac Phospholipids

The fatty acid composition of membrane phospholipids is presented in Table 3. Diabetes induction and the different fortifications did not modify the proportions of saturated, mono-unsaturated, and polyunsaturated fatty acids (SFAs, MUFAs, and PUFAs). However, they did alter the fatty acid profile of each lipid class. In general, diabetes induction increased all the SFAs except palmitic acid (C16:0) which decreased compared to the CTRL group (−29% in general, *p* < 0.001). Similarly, the majority of MUFAs increased, with the exception of C16:1 ω7 and C18:1 ω7 which decreased (−87% and −77% in general, *p* < 0.01 and *p* < 0.001). Diabetes mellitus also increased the ω6/ ω3 PUFA ratio, mainly by augmenting C18:2 ω6 (+31%, *p* < 0.05) and reducing C22:6 ω3 (−54%, *p* < 0.001). GTE enrichment did not modulate the impacts of diabetes. In contrast, EPA enrichment in the context of diabetes and GTE addition limited the proportions of long-chain ω6 PUFAs C22:4 ω6 and C22:4 ω6 (−52% and −82%, *p* < 0.05 and *p* < 0.01, respectively) in favor of membrane EPA (+3.9% in general, *p* < 0.01). 

### 3.4. Morphological Data and Food Intake

During the first three weeks of feeding, all the animals gradually increased their body mass (Figure 3A). This was particularly noticeable amongst rats fed with an HFD, whose fat masses increased markedly more than those of the CTRL rodents (+45% in general, *p* < 0.01, Figure 3B). Consequently, the part of lean mass was reduced in HFD-fed rats (−8%, *p* < 0.05, Figure 3C). 

After the 21st day, weight gain was maintained in the CTRL group until the end of the study, unlike animals injected with STZ. This groups’ body weight fell (−15% in general compared to day 21, *p* < 0.001), and their fat mass decreased (−32% in general compared to day 21, *p* < 0.001). Despite the body weight loss, the DIAB+EPA+GTE group’s food intake increased following the STZ injection, eventually surpassing the intake of the CTRL group at the end of the experiment (+126%, *p* < 0.05).

### 3.5. Plasma Biochemical Parameters

Plasma sodium and chloride concentrations decreased in all diabetic animals (−6 and −9%, *p* < 0.01 and *p* < 0.001, Figure 4A). This was also true for creatinine concentration (−21%, *p* < 0.001, Figure 4B).

Diabetes altered the plasma lipid profile (Figure 4C). In general, the pathology increased plasma triglycerides, total cholesterol, LDL-cholesterol, and non-esterified fatty acid concentrations compared to the CTRL group (+1276, +63, +337, and +794%, *p* < 0.01, *p* < 0.05, *p* < 0.001, and *p* < 0.001, respectively). However, this was not verified for the GTE group which did not differ significantly from the CTRL group for plasma triglycerides and total cholesterol. As shown by a Mann-Whitney test, the GTE enrichment also reduced the plasma NEFA concentration compared to the DIAB group (−59%, *p* < 0.05).

### 3.6. Cardiac Morphology and Mechanical Function

Diabetes increased cardiac weight (+23% in general against the CTRL group, *p* < 0.001), and the different enrichments had no impact (Table 4).

Myocardial collagen content did not show any significant changes following STZ injection. However, the combination of dietary EPA and GTE reduced this content compared to the DIAB animals (−24%, *p* < 0.05 by a Mann-Whitney test).

Regarding cardiac mechanical activity, diabetes did not lead to significant changes in coronary flow and developed pressure. However, when it was normalized to the coronary flow, the developed pressure was higher in the DIAB+GTE animals compared to the DIAB rats without food fortification (+61%, *p* < 0.05 by a Mann-Whitney test). The perfusion pressure was higher in the DIAB rodents than in those of the CTRL group (+68%, *p* < 0.05), whilst the perfusion pressures of the GTE and EPA+GTE groups were between those of the CTRL and DIAB groups.

### 3.7. Coronary Reactivity

The highest EDD (Figure 5A) was observed in the CTRL group. It was strongly decreased by diabetes (−55% in general at the highest injected Ach dose, *p* < 0.001). Interestingly, GTE enrichment associated with EPA fortification increased EDD compared to the diabetic group without enrichment (+167% when the injected Ach dose was 40 pmoles, *p* < 0.001). The diabetes-induced decrease in EDD was mainly due to a decrease in ECVA (Figure 5B) which was close to zero in all of the diabetic groups, but also to a reduction of EID (−35% in general at the highest injected SN dose, *p* < 0.05, Figure 5C). The improved EDD in the DIAB+EPA+GTE group was related to an improved EID (+54% compared to the DIAB group when 500 pmoles of SN was injected, *p* < 0.05), but also to an enhancement of ECVA (+302% compared to the DIAB+GTE group when the injected Ach dose was 40 pmoles, *p* < 0.05). Conversely, GTE enrichment alone did not induce a significant improvement of EDD compared to the non-fortified animals. 

### 3.8. Inflammation of Myocardial Tissue

Myocardial inflammation was first investigated by examining the activation of the NF-κB pathway. IκBα is known to inhibit activation of the NF-κB pathway, and the DIAB group displayed lower levels of this protein than the CTRL group (−56%, *p* < 0.001, Figure 6B). In contrast, GTE and EPA+GTE enrichments restored IκBα to levels close to that of the CTRL group. Despite the observed modifications of IκBα, diabetes induction and dietary fortifications did not change mRNA expression of the pro-inflammatory interleukins targeted in the study (IL-1β, 6, and 10, Figure 6A). 

### 3.9. Cardiac Oxidative Stress

Diabetes induction altered neither the gene expressions of catalase mRNA level, a cytosolic H_2_O_2_ scavenger, nor SOD2, a mitochondrial superoxide detoxifier (Figure 7). In contrast, when both EPA and GTE were added to the diet, increased catalase mRNA expression was found compared to the situations with or without GTE fortification (+107% and +96%, *p* < 0.001). Diabetic rats fortified with GTE displayed reduced SOD2 mRNA expression compared to healthy animals with the normolipidic diet (−52%, *p* < 0.05). However, the rodents fed EPA associated with GTE ceased presenting this difference.

In the mitochondria, protein carbonylation was increased in all the diabetic groups (+130% in general compared to the CTRL group, *p* < 0.01, Figure 7). Protein nitrosylation was reduced by diabetes induction (−32%, *p* < 0.05), but restored with GTE and EPA+GTE enrichments. SOD1 and GPX1 expressions were unaffected by the different manipulations. Similarly, UCP3 and SIRT3 expressions were unchanged.

In the whole myocardium (Figure 7), NOX1 expression was unchanged by diabetes except in the GTE-enriched group where it was reduced (−56%, *p* < 0.05). No variations of SOD1 and SOD2 levels were found but diabetes led to a rise of SOD2 acetylation in DIAB and DIAB+GTE groups (+44% and +35%, respectively, *p* < 0.001). The acetylation was even higher in the DIAB+EPA+GTE group compared to the other two diabetic groups (+46% and +56%, *p* < 0.001). The SOD2 hyperacetylation led to an increased Ac-SOD2/total SOD2 ratio in the DIAB+GTE and DIAB+EPA+GTE groups compared to the CTRL one (+31% and +34%, respectively, *p* < 0.05). In the DIAB group without fortification, the ratio was not significantly increased. No difference was observed in the levels of myocardial GPX1 and SIRT1 proteins.

### 3.10. Energy Metabolism and Mitochondrial Pathways-Related Factors

Lysine acetylation of cardiac and mitochondrial proteins reflects the β-oxidation intensity. It was higher in the diabetic rats, particularly when their diet was enriched with GTE (+197% and +337% for the DIAB+GTE and DIAB+EPA+GTE groups, *p* < 0.05, Figure 8B,C). This occurred while CPT1 expression was unchanged.

Concerning glucose oxidation, the expression of the PDK4 protein, known to inhibit pyruvate conversion to acetylCoA, was increased by diabetes induction (+415% in general, *p* < 0.01, Figure 8B).

Finally, NCX1 and NHE1 connect metabolic acidosis with intracellular calcium, cardiac mechanical activity, and cellular death. Their gene expressions were decreased in the DIAB and DIAB+GTE groups (−38% and −31% in general, *p* < 0.01), but restored close to the CTRL value by the EPA+GTE enrichment (Figure 8A).

## 4. Discussion

This study set out (i) to determine the impact of diabetes induced by the association of STZ injection with a HFD on cardiac function; (ii) to evaluate the effects of enrichments with EPA, GTE, and a combination of both on the consequences observed in this model; and (iii) to examine the mechanisms involved.

### 4.1. Experimental Model

A high dose or several low doses of STZ leads to type I diabetes [46]. In this study, we chose a single low-dose of STZ combined with a HFD to try to trigger type II diabetes (T2D). Several studies have shown that STZ-induced diabetes in combination with a HFD induces either insulin resistance or T2D (early or even late T2D) [47]. However, diabetes characterization was not always performed [48]. In all cases, the treatment always triggered diabetes mellitus. In our study, the glycemia and insulinemia after STZ injection confirmed diabetes induction, but we do not know if it is late T2D or a type I form, in view of the insulin deficiency. Indeed, we were unable to determine the homeostatic model accessment of insulin resistance (HOMA) index, since animal weakness did not allow starvation for several hours.

### 4.2. Animal Morphology and Survival Rate during Diabetes

HFD was provided to create an overweight condition with fat mass accumulation before the STZ injection [49,50]. Our data indicate that the HFD-induced body weight gain was mainly due to fat mass accumulation during this period. However, after diabetes induction the animals displayed a loss of fat mass compared to healthy animals. This decrease was particularly severe in the DIAB+EPA+GTE group. This is surprising since the animals of this group consumed the highest quantity of calories over the post-injection period. Energy wasting probably occurred in this group. Moreover, the lean mass was also decreased. The variation ranged from −5.4 ± 3.9 in the DIAB group to −9.0 ± 4.4% in the DIAB+EPA+GTE one. Diabetic hyperglycemia with insulin deficiency is known to promote hyperosmolarity leading to plasmolysis and therefore tissue dehydration. The phenomenon is characterized by hypochloremia and hyponatremia in the case of diabetic acidosis [51]. Low levels of plasma sodium and chloride ions were observed in our study indicating that dehydration contributed to the loss of lean mass. Furthermore, this was also associated with a decreased plasma creatinine concentration which suggests muscle atrophy and sarcopenia.

The morphological data also highlight the deleterious effect of EPA during diabetes mellitus. Indeed, the DIAB+EPA rats displayed a radical increase in mortality, occurring a few days after the STZ injection, and accompanied with a severe drop in body weight (see Figure 3A). Dehydration can partly explain this loss, since these animals showed a clear decrease in skin elasticity. In 2015, Li et al. published a meta-analysis outlining the deleterious impact of ω3 PUFAs on the treatment of diabetes in caucasian patients [52]. The authors explained the phenomenon by the need to associate antioxidants with omega 3 PUFAs for the treatment of pathologies associated with high oxidative stress [53,54]. EPA is known to be highly susceptible to lipid peroxidation [55]. Our study reaches the same conclusion by showing the protective effect of GTE as an antioxidant in association with EPA. The simple combination of EPA with GTE did indeed allow the drop in body weight to be partially prevented, and thus improve the survival rate.

### 4.3. Cardiac Effects of the Various Interventions

#### Physiological Effects

The diabetes mellitus induced in the present study triggered cardiac hypertrophy irrespective of dietary fortifications. Diabetes in general, and especially type II diabetes, causes hypertension in humans, and this has also been proven in rats [56,57]. Hypertension in turn leads to cardiac hypertrophy [58,59,60]. 

In our study, cardiac mechanical activity was assessed ex vivo by using a model of isolated heart perfused at constant coronary flow. By providing similar amounts of oxygen and substrates to the cells, this model allows the estimation of cardiomyocyte health. No significant differences were noticed except a GTE-related activation of cardiomyocytes in comparison with the diabetic situation without fortification. This can reflect an increased intracellular calcium concentration. Perfusion at constant flow does not reflect the in vivo situation in which the coronary flow is fixed according to the preload pressure. This is the reason why we determined the coronary reactivity, which was greatly decreased by diabetes, slightly improved by GTE and significantly enhanced by the combination of EPA and GTE. Increased coronary reactivity supports increased vascular irrigation when the heart is perfused at constant pressure. This suggests that the in vivo hearts of the DIAB+EPA+GTE rats were better perfused than both those of the DIAB rodents and those of the DIAB+GTE rats receiving intermediary quantities of blood. Since the heart acts as an engine, increasing perfusion improves contraction. Indeed, in a previous ex vivo study we noticed that doubling the coronary flow doubled the developed pressure. In vivo, contractile strength should be the highest in the DIAB+EPA+GTE group and the lowest in the DIAB one. These results fit well with the rate of fibrosis estimated as myocardial collagen. Indeed, cellular death and fibrosis levels were lowest in the DIAB+EPA+GTE group and highest in the DIAB group. Their levels were moderate in the DIAB+GTE group, but the developed pressure was activated: Calcium spikes were probably increased, but not enough to induce a high rate of cellular death. Our results indicate that cardiac health can be sequenced as follows: CTRL > DIAB+EPA+GTE > DIAB+GTE > DIAB. Oxidative stress, membrane lipid composition and metabolism were evaluated in order to understand the mechanisms of the observed effects on the heart.

### 4.4. Oxidative Stress

Oxidative stress plays a fundamental role in diabetes. In our study, the pathology led to high levels of mitochondrial protein carbonylation as described in the literature [61,62]. This was due to the high rate of SOD2 acetylation, which meant that superoxide anions were not detoxified to H_2_O_2_, and thus caused huge molecular damages to the mitochondria. Nitrosylation of mitochondrial proteins was reduced in the DIAB group compared to the CTRL one and restored above control in the two fortified groups. This reaction results from the interaction of nitric oxide (NO) and superoxide anions, which leads to the formation of peroxynitrites which react with tyrosine residues of cellular proteins. Our results show that NO formation was probably low in the high oxidative context of the DIAB group. In diabetes, NO synthase is uncoupled due to the decrease in the redox potential. This increases superoxide anion formation at the expense of NO [63,64]. In contrast, NO synthesis was probably high in the two fortified groups. This suggests lower oxidative stress. The similar rates of protein carbonylation in the three hyperglycemic groups might have resulted from a saturation of the sites susceptible to carbonylation. The information concerning the oxidative stress [(DIAB > (DIAB+GTE ≈ DIAB+EPA+GTE) > CTRL] and NO synthase activity [(DIAB+GTE ≈ DIAB+EPA+GTE) > DIAB] are confirmed by the inflammation status. During diabetes, the oxidative stress promotes inflammation by activating the NF-κB pathway through IκBα release and degradation. IκBα was decreased in the DIAB group while the intake of GTE, with or without EPA, prevented this decrease and therefore limited inflammation.

Changes in oxidative stress and NO availability can play a fundamental role in coronary reactivity, which depends on two types of vascular cells: Endothelial and smooth muscle cells. Endothelial cells release vasoactive agents (NO, prostacyclin, and thromboxane) which regulate the tonus of smooth muscle cells. In this work, as evidenced by the changes in EID and EDD, both types of vascular cells were damaged. An oxidative stress-related decrease in NO availability in the three diabetic groups perfectly explains the reduced ECVA compared to that of the CTRL group. A lower NADPH oxidase 1 myocardial expression was shown with GTE enrichment. NOX1 is expressed in endothelial, smooth muscle, and adventitial cells of the vasculature. It promotes endothelial dysfunction and hypertrophy of smooth muscle cells [65] through release of superoxide anions. Its activity is stimulated in cases of hyperglycemia and hyperlipidemia [66]. The lipid-lowering effect of GTE has already been described [67,68], and it was confirmed in our study in the diabetic situation. As a consequence, NOX1 protein expression reduced, thus protecting both vascular cells and coronary reactivity. Moreover, we noticed a slight improvement of ECVA with EPA+GTE compared to those of the other two diabetic groups. Other vasoactive agents such as prostaglandins and leukotrienes may be responsible for the improved ECVA.

### 4.5. Fatty Acid Composition of Cardiac Phospholipids

Diabetes promoted the accumulation of long-chain SFAs and C18:2 n-6 while reducing the proportion of docosahexaenoic acid (DHA or C22:6 ω3) and increasing the ω6/ω3 ratio. 

The influence of HFDs on cardiac function in non-diabetic subjects has been extensively studied by our group and other investigators. We observed an increased proportion of arachidonic acid in cardiac phospholipids at the expense of C18:2 ω6 [49]. This suggests increased release of cyclooxygenase and lipoxygenase products. The changes in membrane lipid composition were associated with improved coronary reactivity, cardiac mechanical function, and mitochondrial energy metabolism [69], as well as reduced myocardial oxidative stress [54]. We confirmed the influence of abdominal obesity on the function of human atrial mitochondria (publication in process). In the present study, we associated STZ-induced hyperglycemia with HFDs and found opposite results. Indeed, the conversion from C18:2 ω6 to C20:4 ω6 decreased, suggesting reduced prostaglandin and leukotriene synthesis. This was associated with a deteriorated coronary reactivity. The reduced fluxes maybe a reaction of the heart to an excessive production of harmful vasoconstrictors synthesized from PUFAs in diabetic situations (e.g., thromboxane A2). Excess cellular calcium and/or oxidative stress may have shifted eicosanoid synthesis from vasodilators towards vasoconstrictors. 

In contrast, dietary EPA reduced the proportion of long-chain ω6 PUFAs (C22:4 ω6 and C22:5 ω6) and increased that of EPA. This is known to be beneficial for the heart [26,70]. Variations in LC-PUFAs of membrane phospholipids may impact cardiac vascular reactivity due to the vasoactive properties of the metabolites formed from ω3 and ω6 fatty acids [71,72]. An improved vascular reactivity due to ω3 PUFAs has been observed in type II diabetes [73]. This is consistent with our results since we observed an improved EDD in the DIAB+EPA+GTE group compared to the DIAB group.

### 4.6. Energy Metabolism

In addition to its lipid-lowering properties, green tea also protects against lipid peroxidation [74], thus promoting the β-oxidation of fatty acids and providing the optimal substrate for mitochondria in this diabetes model. This avoids energy deficiency and the onset of heart failure. The rate of acetylated-lysine protein in myocardial tissue and mitochondria noticed in our study corresponded with a higher rate of β-oxidation in the GTE-enriched groups and with the results of other investigators [75]. This may be due to green tea EGCG, which is known to improve lipid transport and fatty acid β-oxidation in obesity and type II diabetes [76]. Alongside this, an increased protein expression of PDK4, an inhibitor of the pyruvate dehydrogenase, was observed in all diabetic animals. This increase indicates an inhibition of glucose oxidation. The shift from glucose oxidation toward fatty acid degradation is typical of myocardial insulin resistance and suggests that our STZ-treated and HFD-fed rodents had reached late T2D. In the longer term, this could harm cardiac health, as favoring lipid degradation at the expense of glucose oxidation is detrimental [77,78].

Some of our experimental groups did not display complete β-oxidation because diabetes mellitus provokes ketoacidosis. This results in cellular acidosis, which can be reinforced by excess anaerobic glycolysis, since pyruvate oxidation was partially blocked by diabetes. We measured NHE1 and NCX1 gene expressions in the present study. In physiological conditions, NCX1 contributes to calcium washout from the cardiomyocytes. However, cellular acidosis couples NCX1 activity to that of NHE1, and proton exclusion favors increased cellular calcium, activation of the contractile machinery, and apoptosis in paroxysmal situations [79]. NHE1 and NCX1 gene expressions decreased in the DIAB and DIAB+GTE groups, suggesting that the cardiac cells prevented excessive activation of this axis because of acidosis. Inhibition of NHE1 has a cardio-protective effect by limiting NCX1-related calcium overload [80]. It has been shown that green tea epicatechin inhibits NHE1 in erythrocytes [81]. This probably limited excess intracellular calcium in the DIAB+GTE rats compared to the DIAB rodents. Cardiac pressure thus increased since calcium concentration remained high, but not high enough to induce cellular death. Moreover, as suggested by the NHE1 and NCX1 gene expressions, acidosis was not observed in the EPA+GTE-fortified rodents. Alongside this, coronary reactivity was quasi-normal and fibrosis was low. This suggests that calcium did not accumulate in cardiac cells in spite of the high degree of mitochondrial oxidative stress.

## 5. Conclusions

The combination of HFDs with STZ-induced diabetes in rodents triggered diabetes mellitus, probably corresponding to late T2D. This slightly decreased the survival rate of the animals. Most importantly, addition of EPA to the diet drastically increased the rate of mortality due to dehydration and catabolism. However, this phenomenon was counterbalanced by dietary GTE, probably because of its antioxidant properties. The rate of mortality was not correlated with cardiac function, since large variations in myocardial activities were noticed although no significant changes in survival rates were observed between the DIAB, DIAB+GTE and DIAB+EPA+GTE groups. Diabetes-induced deteriorations in cardiac function were related to the oxidative stress and modulation of cellular acidosis associated with changes in cellular calcium handling. Dietary GTE fortification improved cardiac function through two mechanisms: Improved NO production and probable epicatechin-related inhibition of NHE1. Further addition of EPA in the diet mainly suppressed acidosis and established a favorable PUFA profile in cardiac phospholipids. Finally, the exact mechanism by which EPA increases the mortality rate in the diabetic situation requires further examination. 

## Figures and Tables

**Figure 1 antioxidants-08-00526-f001:**
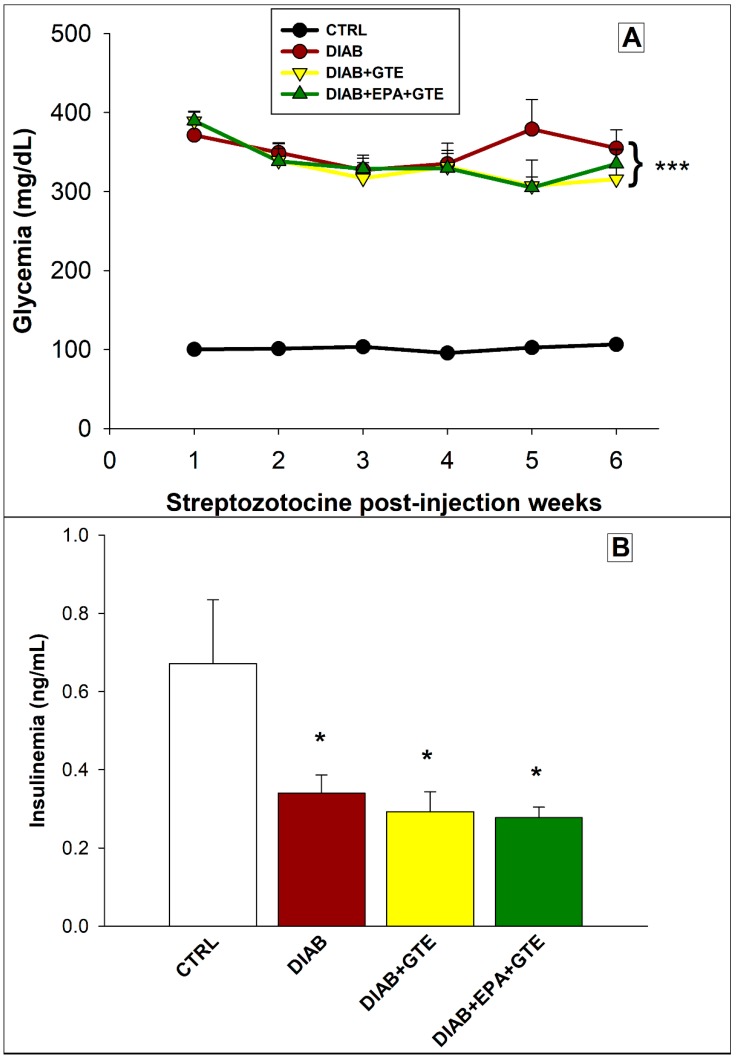
Glycemia and insulinemia. (**A**) Glycemia after the streptozotocin injection and (**B**) Insulinemia on the day of euthanasia. CTRL: Control group; DIAB: Diabetic rats; GTE: Green tea extract enrichment; and EPA: Eicosapentaenoic acid enrichment. *: Different to the CTRL group. One symbol: *p* < 0.05; two symbols: *p* < 0.01; and three symbols: *p* < 0.001.

**Figure 2 antioxidants-08-00526-f002:**
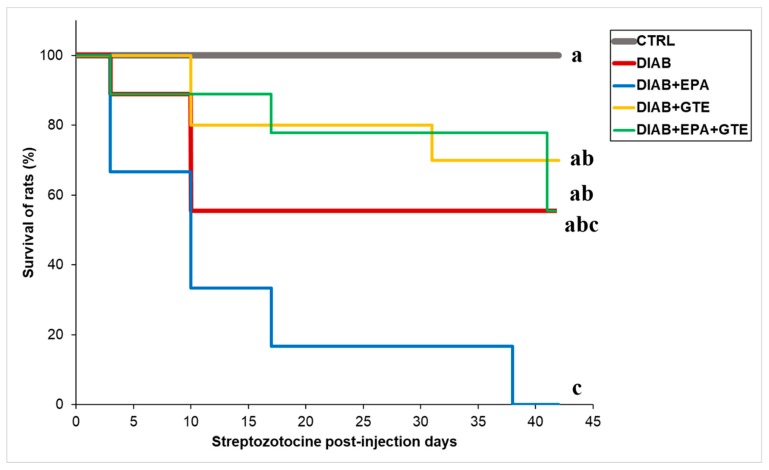
Survival rate of the animals. CTRL: Control group; DIAB: Diabetic rats; GTE: Green tea extract enrichment; and EPA: Eicosapentaenoic acid enrichment. a, b, c: means without a common letter on each line are significantly different, *p* < 0.05.

**Figure 3 antioxidants-08-00526-f003:**
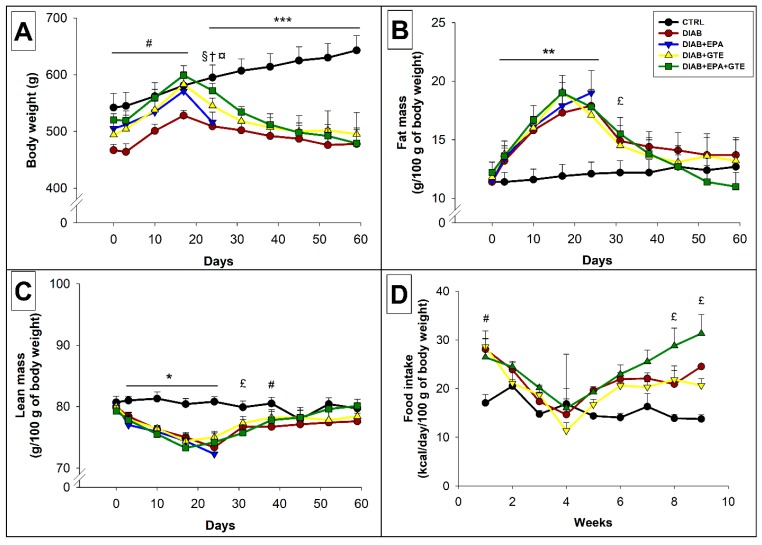
Morphological data and food intake. (**A**) Evolution of the body weight in the different groups; (**B**) Evolution of the fat mass in the different group; (**C**) Evolution of the lean mass in the different group; and (**D**) Rat food intake over the course of the study. CTRL: Control group; DIAB: Diabetic rats; GTE: Green tea extract enrichment; and EPA: Eicosapentaenoic acid enrichment. *: CTRL group different to the others groups; #: CTRL group different to the DIAB group; £: CTRL group different to the DIAB+EPA+GTE group; †: DIAB group different to the DIAB+GTE group; ¤: DIAB group different to the DIAB+EPA+GTE group; and §: DIAB+GTE group different to the DIAB+EPA+GTE group. One symbol: *p* < 0.05; two symbols: *p* < 0.01; and three symbols: *p* < 0.001.

**Figure 4 antioxidants-08-00526-f004:**
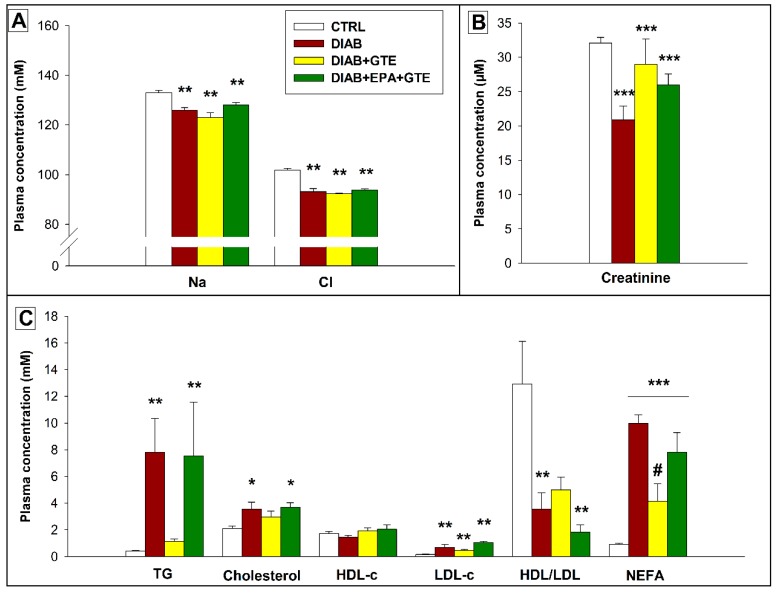
Plasma biochemical parameters. (**A**) Plasma ion diagram; (**B**) Plasma creatinine levels; and (**C**) Plasma lipid profiles. CTRL: Control group; DIAB: Diabetic rats; GTE: Green tea extract enrichment; and EPA: Eicosapentaenoic acid enrichment. Na: Sodium concentration; Cl: Chloride concentration; TG: Triglycerides; HDL-c: HDL-cholesterol; LDL-c: LDL-cholesterol; HDL/LDL: HDL-cholesterol to LDL-cholesterol ratio; and NEFA: Non-esterified fatty acids. * Different to the CTRL group. # Different to the DIAB group. One symbol (*): *p* < 0.05; two symbols (**): *p* < 0.01; and three symbols (***): *p* < 0.001.

**Figure 5 antioxidants-08-00526-f005:**
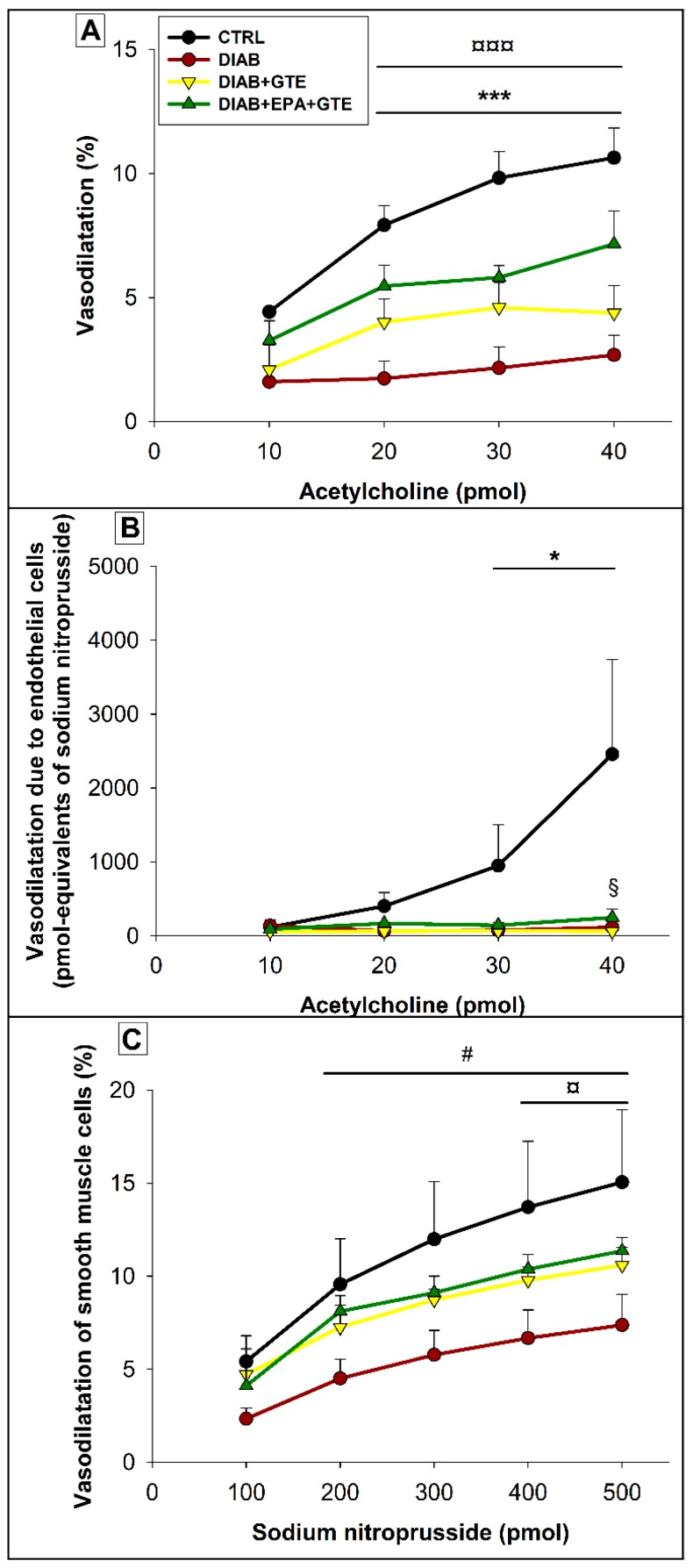
Coronary reactivity. (**A**) Endothelial-dependent vasodilatation due to the combined activities of endothelial and smooth muscle; (**B**) Endothelial cell vasodilation capacities; and (**C**) Endothelial-independent vasodilatation due the smooth muscle cells. CTRL: Control group; DIAB: Diabetic rats; GTE: Green tea extract enrichment; and EPA: Eicosapentaenoic acid enrichment. * CTRL group different to the others groups; #: CTRL group different to the DIAB group; ¤: DIAB group different to the DIAB+EPA+GTE group; and §: DIAB+GTE group different to the DIAB+EPA+GTE group. One symbol: *p* < 0.05; two symbols: *p* < 0.01; and three symbols: *p* < 0.001.

**Figure 6 antioxidants-08-00526-f006:**
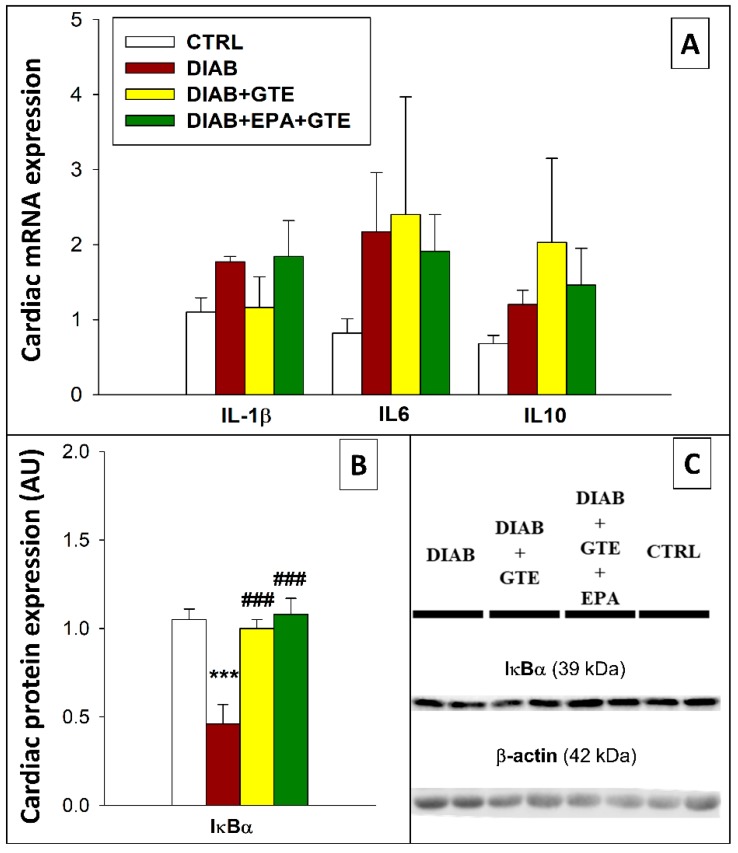
Inflammation parameters. (**A**) Inflammation-related mRNA expression; (**B**) Inflammation-related protein expression; and (**C**) Representative immunoblots. CTRL: Control group; DIAB: Diabetic rats; GTE: Green tea extract enrichment; and EPA: Eicosapentaenoic acid enrichment. *: Different to the CTRL group; #: Different to the DIAB group. One symbol: *p* < 0.05; two symbols: *p* < 0.01; and three symbols: *p* < 0.001.

**Figure 7 antioxidants-08-00526-f007:**
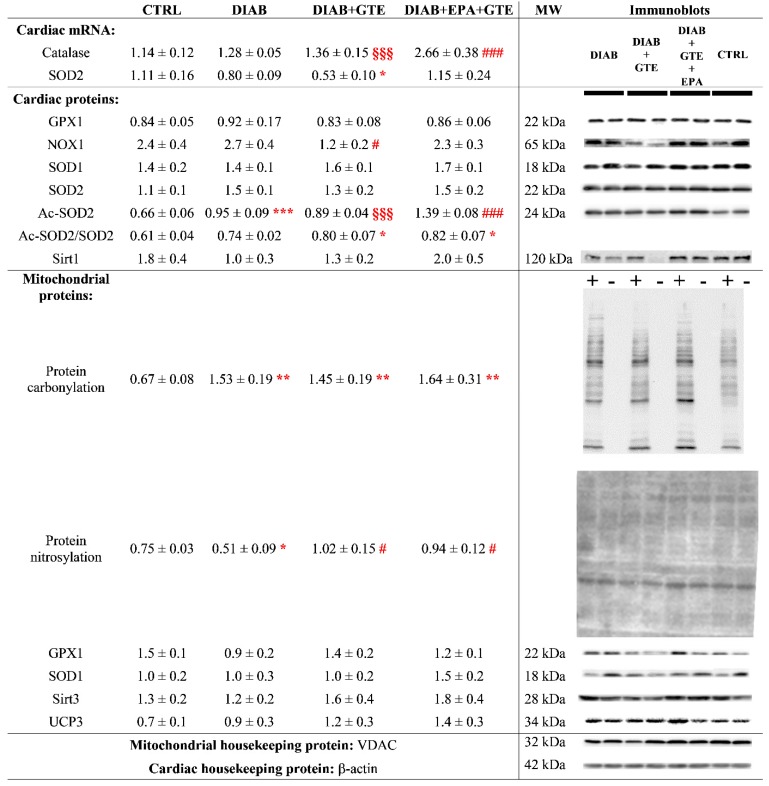
Oxidative stress parameters. CTRL: Control group; DIAB: Diabetic rats; GTE: Green tea extract enrichment; EPA: Eicosapentaenoic acid enrichment; +: Sample processed to detect protein carbonylation; −: Negative control of protein carbonylation; and MW: Molecular weight. *: Different to the CTRL group; #: Different to the DIAB group; §: DIAB+GTE group different to the DIAB+EPA+GTE group. One symbol: *p* < 0.05; two symbols: *p* < 0.01; and three symbols: *p* < 0.001.

**Figure 8 antioxidants-08-00526-f008:**
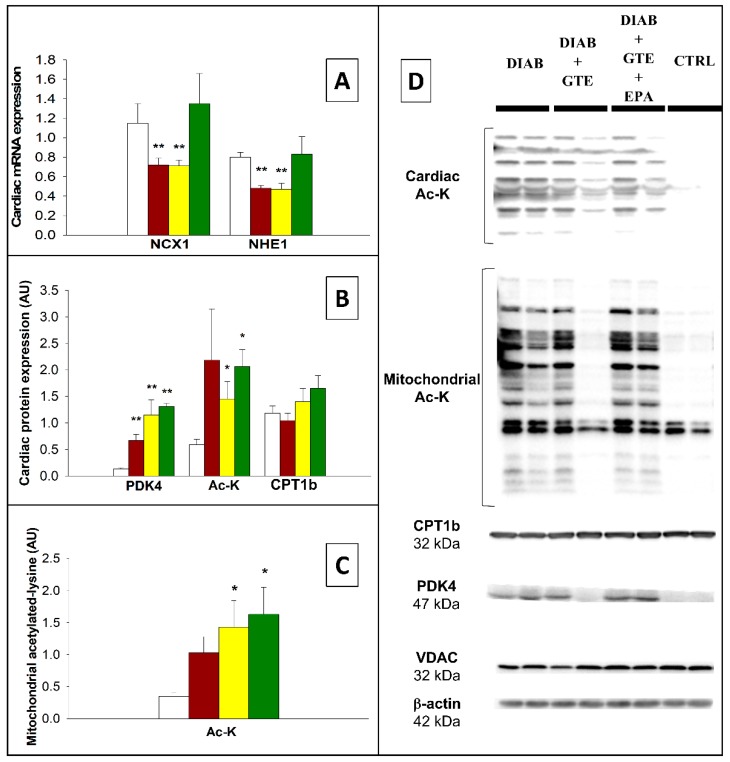
Factors related to the energy metabolism. (**A**) NCX1 and NHE1 mRNA expression; (**B**) Energy metabolism-related myocardial proteins; (**C**) Mitochondrial acetylated-lysine levels; and (**D**) Representative immunoblots. CTRL: Control group; DIAB: Diabetic rats; GTE: Green tea extract enrichment; and EPA: Eicosapentaenoic acid enrichment. *: Different to the CTRL group. One symbol: *p* < 0.05; two symbols: *p* < 0.01; and three symbols: *p* < 0.001.

**Table 1 antioxidants-08-00526-t001:** Composition of the diets.

Food Components	CTRL	DIAB	DIAB+EPA	DIAB+GTE	DIAB+EPA+GTE
**Fatty acid composition:** (% of lipids of the diet)					
C14:0		14.27	12.29	13.36	12.71
C15:0		1.39	1.17	1.24	1.19
C16:0	20.32	38.25	33.78	36.96	34.3
C17:0			0.59	0.63	0.6
C18:0		10.85	10.23	9.81	10.18
C20:0			0.04		
SFAs	20.32	64.76	58.1	62.01	58.97
C14:1		1.65	1.18	0.84	0.75
C16:1 ω7		1.34	1.4	1.78	1.69
C17:1			0.1		0.16
C18:1 trans		1.46	2.13	1.66	2.14
C18:1 ω9	20.43	20.61	22.99	23.61	23.45
C18:1 ω7	1.69		0.73		
MUFAs	22.11	25.05	28.53	27.89	28.19
C18:2 ω6 cis	53.66	8.67	8.74	8.64	8.7
C20:4 ω6			0.25		0.2
ω6 PUFAs	53.66	8.67	8.99	8.64	8.9
C18:3 ω3	3.9	1.52	4.38	1.46	1.51
C18:4 ω3			0.22		
C20:5 ω3			2.67		2.43
ω3 PUFAs	3.9	1.52	4.38	1.46	3.94
ω6/ω3 PUFAs	13.75	5.71	2.05	5.9	2.26
**Lipid part of diet** (%)	3	35.2	35.2	35.2	35.2
**EPA** (% of the diet)	/	/	1.2	/	1.2
**GTE** (% of the diet)	/	/	/	0.5	0.5
**GTE component:** (% of GTE)					
Total phospholipids	/	/	/	≥40	≥40
Catechins	/	/	/	≥25	≥25
EGCG	/	/	/	≥10	≥10
Caffeine	/	/	/	≤7	≤7
**Vitamin E:** (mg/kg of EFM)	/	346	419	357	430
**Peroxide value:** (mEq O_2_/kg of EFM)	/	3	6.9	3.1	5.6

CTRL: Control group; DIAB: Diabetic rats; GTE: Green tea extract; and EPA: Eicosapentaenoic acid. SFAs: Saturated fatty acids; MUFAs: Mono-unsaturated fatty acids; PUFAs: Polyunsaturated fatty acids; EGCG: Epigallocatechin gallate; and EFM: Extracted fat mass.

**Table 2 antioxidants-08-00526-t002:** Real-time PCR primers used in this study.

Gene	Sequence (5′-3′)
**β-Actin** (NM_031144.3)	(F) TCTGTGTGGATTGGTGGCTCTA (R) CTGCTTGCTGATCCACATCTG
**Catalase** (NM_012520.2)	(F) CCTGACATGGTCTGGGACTTC (R) AGCTTGAAGGTGTGTGAGCC
**GAPDH** (NM_017008.4)	(F) GAACATCATCCCTGCATCCA (R) CCAGTGAGCTTCCCGTTCA
**IL10** (NM_012854.2)	(F) GTAGAAGTGATGCCCCAGGC (R) AGACACCTTTGTCTTGGAGC
**IL-1β** (NM_031512.2)	(F) AAATGCCTCGTGCTGTCTGA (R) GGTGTGCCGTCTTTCATCAC
**IL6** (NM_012589.2)	(F) AGCGATGATGCACTGTCAGA (R) GGAACTCCAGAAGACCAGAGC
**NCX1** (NM_019268.3)	(F) TGTTGTTAAATGAGCTTGGTGG (R) TTACGGTAGAGGGAATCGGATG
**NHE1** (NM_012652.1)	(F) AACGGCTGCGGTCCTATAAC (R) CGAGACATGGTGGGTGAGTC
**SOD2** (NM_017051.2)	(F) TGAACAATCTGAACGTCACCG (R) CCTTAGGGCTCAGGTTTGTC

(F): Forward; (R): Reverse. GAPDH: Glyceraldehyde-3-phosphate dehydrogenase; IL-1β: Interleukin-1β; IL-6: Interleukin-6; IL-10: Interleukin-10; NCX1: Sodium-calcium exchanger 1; NHE1: Sodium-proton exchanger 1; and SOD2: Superoxide dismutase 2.

**Table 3 antioxidants-08-00526-t003:** Fatty acids composition of cardiac phospholipids.

Fatty Acid	CTRL	DIAB	DIAB+GTE	DIAB+EPA+GTE	*p*-Value
C12:0	0.44 ± 0.05	0.42 ± 0.02	0.34 ± 0.04	0.65 ± 0.17	NS
C14:0	0.32 ± 0.07 **a**	0.30 ± 0.08 **a**	0.30 ± 0.06 **a**	0.69 ± 0.15 **b**	*p* < 0.05
C15:0	0.23 ± 0.03 **a**	0.30 ± 0.04 **ab**	0.32 ± 0.01 **ab**	0.40 ± 0.04 **b**	*p* < 0.05
C16:0	13.6 ± 0.4 **a**	9.1 ± 0.6 **b**	10.0 ± 0.6 **b**	9.9 ± 0.5 **b**	*p* < 0.001
C17:0	0.30 ± 0.01 **a**	0.36 ± 0.02 **b**	0.39 ± 0.01 **b**	0.39 ± 0.01 **b**	*p* < 0.001
C18:0 DMA	0.31 ± 0.04 **a**	0.61 ± 0.10 **b**	0.67 ± 0.02 **b**	0.73 ± 0.11 **b**	*p* < 0.01
C18:0	19.0 ± 0.3 **a**	23.1 ± 0.2 **b**	23.2 ± 0.2 **b**	22.4 ± 0.3 **b**	*p* < 0.001
SFAs	34.2 ± 0.4	34.2 ± 0.7	34.4 ± 0.4	36.1 ± 0.6	NS
C15:1	0.08 ± 0.01 **a**	0.21 ± 0.03 **b**	0.24 ± 0.03 **b**	0.19 ± 0.05 **b**	*p* < 0.01
C16:1 ω7	0.54 ± 0.05 **a**	0.06 ± 0.02 **b**	0.08 ± 0.01 **b**	0.07 ± 0.01 **b**	*p* < 0.01
C17:1	0.24 ± 0.02	0.20 ± 0.04	0.21 ± 0.02	0.25 ± 0.03	NS
trans-C18:1	0.01 ± 0.01 **a**	0.18 ± 0.00 **b**	0.20 ± 0.01 **b**	0.19 ± 0.02 **b**	*p* < 0.01
C18:1 ω9	4.0 ± 0.2 **a**	6.5 ± 0.6 **b**	6.5 ± 0.5 **b**	7.2 ± 0.6 **b**	*p* < 0.01
C18:1 ω7	5.0 ± 0.2 **a**	1.1 ± 0.1 **b**	1.2 ± 0.0 **b**	1.2 ± 0.1 **b**	*p* < 0.001
C22:1 ω9	0.87 ± 0.02 **a**	0.81 ± 0.08 **ab**	0.86 ± 0.14 **ab**	0.68 ± 0.06 **b**	*p* < 0.05
MUFAs	10.8 ± 0.4	9.2 ± 0.5	9.4 ± 0.4	10.3 ± 0.5	NS
C18:2 ω6	23.5 ± 0.3 **a**	30.9 ± 2.9 **b**	30.1 ± 1.1 **b**	30.1 ± 2.3 **b**	*p* < 0.05
C20:2 ω6	0.04 ± 0.02	0.00 ± 0.00	0.03 ± 0.02	0.09 ± 0.06	NS
C20:3 ω6	0.20 ± 0.02 **a**	0.55 ± 0.01 **b**	0.50 ± 0.02 **b**	0.43 ± 0.02 **c**	*p* < 0.001
C20:4 ω6	18.2 ± 0.4 **a**	17.8 ± 1.7 **ab**	18.6 ± 1.0 **ab**	15.7 ± 1.1 **b**	*p* < 0.05
C22:4 ω6	0.66 ± 0.04 **a**	0.58 ± 0.04 **a**	0.58 ± 0.06 **a**	0.28 ± 0.02 **b**	*p* < 0.05
C22:5 ω6	0.77 ± 0.07 **a**	0.50 ± 0.12 **ab**	0.41 ± 0.09 **bc**	0.09 ± 0.02 **c**	*p* < 0.01
ω6PUFAs	43.6 ± 0.6 **a**	50.4 ± 1.6 **b**	47.8 ± 1.7 **b**	48.3 ± 0.6 **b**	*p* < 0.01
C20:5 ω3	0.01 ± 0.01 **a**	0.03 ± 0.02 **a**	0.05 ± 0.02 **a**	1.21 ± 0.12 **b**	*p* < 0.01
C22:5 ω3	1.19 ± 0.10 **a**	1.58 ± 0.26 **ab**	2.21 ± 0.41 **b**	3.72 ± 0.97 **b**	*p* < 0.05
C22:6 ω3	10.2 ± 0.57 **a**	4.7 ± 1.0 **b**	4.0 ± 0.2 **b**	3.3 ± 0.4 **b**	*p* < 0.001
ω3 PUFAs	11.4 ± 0.6 **a**	6.3 ± 1.2 **b**	6.2 ± 0.5 **b**	7.7 ± 0.2 **ab**	*p* < 0.01
PUFAs	55.0 ± 0.5	56.6 ± 0.7	55.5 ± 0.6	55.5 ± 0.4	NS
ω6/ω3 PUFAs	3.9 ± 0.3 **a**	9.8 ± 2.3 **b**	7.4 ± 1.1 **b**	5.5 ± 0.7 **ab**	*p* < 0.05

CTRL: Control group; DIAB: Diabetic rats; GTE: Green tea extract; and EPA: Eicosapentaenoic acid. SFAs: Saturated fatty acids; MUFAs: Mono-unsaturated fatty acids; and PUFAs: Polyunsaturated fatty acids. a, b, c: means without a common letter on a same line are significantly different.

**Table 4 antioxidants-08-00526-t004:** Cardiac morphological and functional parameters.

Cardiac Parameters	CTRL	DIAB	DIAB+GTE	DIAB+EPA+GTE
**Heart mass** (g/100 g of body weight)	0.26 ± 0.01	0.32 ± 0.01 ***	0.33 ± 0.01 ***	0.30 ± 0.01 ***
**Collagen** (µg/mg of proteins)	163 ± 5	170 ± 15	154 ± 24	129 ± 6 #
**Coronary flow** (mL/min/g of heart)	8.29 ± 0.38	9.40 ± 0.46	8.87 ± 0.76	9.38 ± 0.21
**Developed pressure** (mmHg)	66.7 ± 4.2	51.3 ± 10.3	73.0 ± 4.3	66.2 ± 10.1
**Developed pressure/coronary flow** (mmHg/mL/min/g of heart)	8.03 ± 0.34	5.76 ± 1.45	9.29 ± 1.17 #	7.03 ± 1.00
**Heart rate** (bpm)	368.0 ± 0.3	368.6 ± 0.2	368.6 ± 0.4	368.4 ± 0.2
**RPP** (bpm·mmHg)	24425 ± 1534	19021 ± 3795	26947 ± 1580	24377 ± 3721
**dPdtmax** (mmHg·s^−1^)	2458 ± 341	1532 ± 310	2067 ± 356	1988 ± 322
**dPdtmin** (mmHg·s^−1^)	−1497 ± 119	−1277 ± 280	−1749 ± 98	−1565 ± 201
**Perfusion pressure** (mmHg)	50.4 ± 6.0	84.8 ± 11.1 *	70.4 ± 5.7	67.04 ± 10.8

CTRL: Control group; DIAB: Diabetic rats; GTE: Green tea extract enrichment; and EPA: Eicosapentaenoic acid enrichment. *: Different to the CTRL group; #: Different to the DIAB group. One symbol (* or #): *p* < 0.05; two symbols (**): *p* < 0.01; and three symbols (***): *p* < 0.001.

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
