# Peer review of "Dietary EPA Increases Rat Mortality in Diabetes Mellitus, a Phenomenon Which Is Compensated by Green Tea Extract"

_antioxidants, 2019, doi:10.3390/antiox8110526_

Round 1

Reviewer 1 Report

Leger and co-workers did really a great job, the results are correctly supported by a great amount of data, all the studies are well conducted and the manuscript is well written. I really appreciated it, i Suggest to accept it after minor language check.

Author Response

All the authors of this manuscript thank the reviewer for its exciting evaluation. Corrections related to the English language have been performed.

Reviewer 2 Report

Line 78 You should include another paper with green tea and STZ diabetes which shows multiple improvements in mechanisms leading to diabetic complications (Vinson, 2005).

Line 81 reference 35 doesn't examine these properties and should be deleted

line 188 how was 13 mg/ml determined?  Is this protein concentration?

Line 195, 196 protein was not proteins were

Line 213 For detection ... should be a separate paragraph

Line 241 three times a week not thrice

line 276 However mortality occurred in all ...

Line 283 move over to margin

Line 334 chloride not chlorine

Table 4 cardiac parameters should be bold and units regular font

Figure 5 A are the three symbols for data between 20 and 40 min?  If the answer is yes then the figure is correct

Line 408 move to margin.  found not noticed

line 419 found not noticed

genic should be replaced with gene ?

Line 591 remove developed

Author Response

All the authors of this manuscript thank the reviewer for its acute evaluation. Corrections related to the English language have been performed.

The following changes were made:

Line 78 You should include another paper with green tea and STZ diabetes which shows multiple improvements in mechanisms leading to diabetic complications (Vinson, 2005).

The publication of Vinson and Zhang (2005) has been added and three other references explaining green tea hypoglycemic effect during diabetes have been added.

Line 81 reference 35 doesn't examine these properties and should be deleted

Reference [35] has been discarded and replaced by the publication authored by Winson et al (2004).

line 188 how was 13 mg/ml determined?  Is this protein concentration?

13 mg/ml is the averaged concentration we obtained after protein determination by the bicinconinic acid technique. The true unit has been detailed.

Line 195, 196 protein was not proteins were

“proteins were” has been replaced by “protein was”

Line 213 For detection ... should be a separate paragraph

A separate paragraph has been created for detection of the antibodies.

Line 241 three times a week not thrice

“thrice a week” has been replaced by three times a week”.

line 276 However mortality occurred in all ...

“However, the mortality was extended in all” has been replaced by “However, mortality occurred in all”.

Line 283 move over to margin

The text has been moved over to margin.

Line 334 chloride not chlorine

“chlorine” has been replaced by “chloride”

Table 4 cardiac parameters should be bold and units regular font

Cardiac parameters have been written in gold and the units in regular font.

Figure 5 A are the three symbols for data between 20 and 40 min?  If the answer is yes then the figure is correct

The three symbols are for the data between 20 and 40 pmoles.

Line 408 move to margin.  found not noticed

The text has been removed to margin. “noticed” was replaced by “found”.

Line 419 found not noticed

“noticed” has been replaced by “found”.

genic should be replaced with gene ?

“genic” has been replaced by “gene” all along the text.

Line 591 remove developed

“developed” has been suppressed.